# Effect of a loss of the *mda5/ifih1* gene on the antiviral resistance in a Chinook salmon *Oncorhynchus tshawytscha* cell line

**Catherine Collins**[1,2☯], **Lise Chaumont**[1☯], **Mathilde Peruzzi**[1], **Nedim Jamak**[1],
**Pierre Boudinot**[1], **Julia Béjar**[3], **Patricia Moreno**[3], **Daniel Álvarez Torres**[3],
**Bertrand Collet**[1]*

**1** INRAE, UVSQ, VIM, Université Paris-Saclay, Jouy-en-Josas, France, **2** School of Biological, Earth and Environmental Sciences, University College Cork, Cork, Ireland, **3** IBYDA, Universidad de Málaga, Málaga, Spain

☯ These authors contributed equally to this work.
* bertrand.collet@inrae.fr

**Data Availability Statement:** All relevant data are within the manuscript and its Supporting Information files.

## Abstract

Cells are equipped with intracellular RIG-like Receptors (RLRs) detecting double stranded (ds)RNA, a molecule with Pathogen-Associated Molecular Pattern (PAMPs) generated during the life cycle of many viruses. Melanoma Differentiation-Associated protein 5 (MDA5), a helicase enzyme member of the RLRs encoded by the *ifih1* gene, binds to long dsRNA molecules during a viral infection and initiates production of type I interferon (IFN1) which orchestrates the antiviral response. In order to understand the contribution of MDA5 to viral resistance in fish cells, we have isolated a clonal Chinook salmon *Oncorhynchus tshawytscha* epithelial-like cell line invalidated for the *ifih1* gene by CRISPR/Cas9 genome editing. We demonstrated that IFN1 induction is impaired in this cell line after infection with the Snakehead Rhabdovirus (SHRV), the Salmon Alphavirus (SAV) or Nervous Necrosis Virus (NNV). The cell line, however, did not show any increase in cytopathic effect when infected with SHRV or SAV. Similarly, no cytopathic effect was observed in the *ifih1*[-/-] cell line when infected with Infectious Pancreatic Necrosis Virus (IPNV), Infectious Haemorrhagic Necrotic Virus (IHNV). These results indicate the redundancy of the antiviral innate defence system in CHSE-derived cells, which helps with circumventing viral evasion strategies.

## Introduction

Vertebrate innate immunity in response to viruses is regulated by the type I interferons (IFN1s). Most nucleated cells are equipped with a number of cellular sensors of viruses, called Pattern Recognition Receptors (PRRs). Upon binding to viral components, these sensors trigger a signalling cascade involving phosphorylation and activation of signalling pathways which results in the transcription of genes coding for IFN1s. Secreted IFN1s, in turn, through specific signalling processes, are responsible for the induction of a large number of Interferon Stimulated Genes (ISGs) leading to the establishment of an antiviral state [1]. This occurs in cells

**Funding:** The funders had no role in study design, data collection and analysis, decision to publish, or preparation of the manuscript.

**Competing interests:** The authors have declared that no competing interests exist.

harbouring specific membrane-bound IFN1s receptors, which in mammals represent the majority of nucleated cells [2]. The general systemic antiviral state effectively prevents viral replication and propagation.

Among the PRRs, Melanoma Differentiation-Associated Protein 5 (MDA5) is a DExD/H box RNA helicase that belongs to the retinoic acid inducible gene-I (RIG-I)-like receptor family along with RIG-I and LGP2. The protein is encoded by the *ifih1* gene and is composed of 2 CARD domains located at the N-terminus, a hinge region and a helicase domain composed of sub-domains RecA-like Hel1 and Hel2. A second hinge region connects the C-terminal domain (CTD) which recognises and binds RNA [3]. Upon viral dsRNA binding, MDA5 triggers the MAVS/TBK1/IRF3/IRF7/NF-κB pathway resulting in the specific induction of genes encoding IFN1s followed by their production and secretion [4].

In fish, the PRR pathways, based on their gene homology, are generally well conserved [5–7]. However, it is unclear whether all categories of viruses are detected by each specific member of the RLRs (MDA5, RIG-I or LPG2). The *ifih1* gene has been identified in a large number of fish species (Grass carp *Ctenopharyngodon Idella* [8], Japanese flounder *Paralichthys olivaceus* [9], Rainbow trout *Oncorhynchus mykiss* [10], Channel catfish *Ictalurus punctatus* [11], Zebrafish *Danio rerio* [12], Green chromide *Etroplus suratensis* [13], Orange spotted grouper *Epinephelus coioides* [14], Sea perch, *Lateolabrax japonicus* [15], Common carp *Cyprinus carpio* [16], Large yellow croaker *Larimichthys crocea* [17], Atlantic salmon *Salmo salar* [18], Black carp *Mylopharyngodon piceus* [19], Asian seabass *Lates calcarifer* [20], Tilapia *Oreochromis niloticus* [21], Mandarin fish *Synchiropus splendidus* [22], Barbel chub *Squaliobarbus curriculus* [23]). These numerous studies showed that the *ifih1* gene is present as a unique copy in the genome of the species investigated, and as in mammals, induction of *ifih1* gene by IFN1s has been demonstrated in fish [24–27].

To determine the contribution of MDA5 to induction of IFN1s response in fish upon viral infection, we have engineered a stable cell line derived from Chinook salmon *Oncorhynchus tshawytscha* with a *ifih1* gene invalidated by targeted mutation using CRISPR/Cas9 genome editing and have characterised its ability to induce IFN1s and to resist to viral infection.

## Material and methods

### Genome analysis and isolation of the CHSE-EC cell line mutated for *ifih1* gene

To confirm the uniqueness of the *ifih1* gene in *Oncorhynchus tshawytscha*, tblastn analysis was used to search the genome (Otsh_v2.0, GCA_018296145.1) using the *Oncorhynchus tshawytscha* interferon-induced helicase C domain-containing protein 1 amino acid sequence (XP_024244677) as a query. The phylogeny of MDA5 was inferred by using the Maximum Likelihood method and the JTT matrix-based model [28] and was conducted in MEGA11 [29]. A single guide RNA (sgRNA) was designed within the first coding exon of the *ifih1* gene (GeneID 112225192, chromosome 26) at the site CGCCAACATAAGTCTGATTG using the CRISPOR software [30]. A dsDNA template was amplified by PCR using the Q5 2X mastermix according to the manufacturer's instruction (New England Biolabs) and using DR274F and R1 primers and the long primer T1 used as the PCR template (Table 1). The resulting PCR product was purified (Macherey-Nagel, PCR and Gel purification kit), used as template for T7 RNA *in vitro* synthesis (Promega Ribomax T7 kit), and the resulting RNA purified using Trizol reagent (Thermofisher), all steps following the manufacturers' instructions. The *megfp* sgRNA [31], used to screen for successful genome editing in CHSE-EC [31], was produced as described above. One μl of each sgRNA (1μg or 31 pmol) was mixed with 1 μl recombinant nCas9n (1μg or 6.13 pmol, Thermofisher TrueCut2.0) separately, incubated at room

**Table 1. DNA oligonucleotides used in this study for the isolation and genotyping of MDA5C1 and MDA5C2 cell lines.**

| Name | Sequence 5'-3' | purpose |
|---|---|---|
| DR274F | AAAAGCACCGACTCGGTGCCAC | Amplification of the sgRNA T7 *in vitro* transcription template |
| R1 | AGCTAATACGACTCACTATATGGACTTCAGGCCTAGGCTG | |
| T1 | AAAAGCACCGACTCGGTGCCACTTTTTCAAGTTGATAACGGACTAGCCTTATTTTAACTTGCTATTTCTAGCTCTAAAAACCAGCGTAGGGCTGAAGTCCA | |
| otifih1gF | CCAGATTGAAGAGGGAGAAACG | Genotyping the at the ifih1 sgRNA location |
| otifih1gR | CAGTTGTCATTCTCTGCCTCC | |

**Table 2. DNA oligonucleotides used for gene expression by qPCR.**

| Gene | Name | Sequence 5'-3' | GeneID/Ref |
|------|------|----------------|------------|
| *β-actin* | bact_F | GTCACCAACTGGGACGACAT[2] | 112240903[2], 112240902[2], 121844290[2] |
| | bact_R | GTACATGGCAGGGGTGTTGA[2] | |
| *ifn1* | sifn1_F | TCATCTGGATAACTAACAGCGAAAC[1] | 112258508 |
| | sifn1_R | TGTGATATCTCCTCCCATCTGGTC[1] | |
| *irf3* | irf3_F | CAAGGCGTGGGCTGAGG | 112235560 |
| | irf3_R | CTGGGTGCTGAGATCCTCCTG | |
| *shrv-g* | shrvg_F | CGAGTACTCGGAAGAATGGG | AF147498 |
| | shrvg_R | GTGAGGCCTAGATTCTGGTC | |
| *sav-e2* | savE2_F | CGTCACCTTCACCAGCGACTCCCAGACG | NC_003433[3] |
| | savE2_R | GGATCCATTCAGATGTGGCGTTGCTATGG | |
| RG965RNA2 | F4 | ACCGTCCGCTGTCTATTGACTA | Unpublished |
| | R1 | CAGATGCCCCAGCGAAACC | |
| SJ-RNA2 | F | GACACCACCGCTCCAATTACTAC | NC_003449.1 |
| | R | ACGAAATCCAGTGTAACCGTTGT | |

[1] from [43], 100% match between *Oncorhynchus mykiss* and *O. tshawytscha*, specific for isoform X1 encoding for the secreted IFN1

[2] [42]; several paralogues with 100% identity on amplicon

[3] E2 gene, from [36]

temperature for 5 min, pooled and used to transfect the mEGFP fluorescent recombinant CHSE-EC cell line as described previously [31]. Briefly, 4 μl Cas9-sgRNAs was mixed with 30 μl of CHSE-EC suspension in Leibovitz's L-15 medium (Thermofisher Gibco) at $10^7$ cells/ml; 3 electroporation cycles were conducted using 10μl of mix each cycle and using the Neon transfection kit and electroporation machine. The CHSE-EC and derivative cell lines were cultivated as described in [31]. All transfected cells were mixed in 7 ml CHSE-EC culture medium in a 25cm$^2$ flask and incubated at 20°C for two weeks, until the cell population reached confluency. The cells were then passaged (surface ratio 1:3), and one third was used for genomic DNA extraction according to the manufacturer's instruction (Macherey-Nagel, Tissue DNA extraction kit, protocol for animal cells). A 515-nucleotide fragment containing the *ifih1* targeted site was amplified by PCR (New England Biolabs Q5 2X mastermix) using the primers otifih1gF and otifih1gR (Table 2), purified (Macherey-Nagel, Gel and PCR purification kit), and sequenced (Sanger sequencing service, Eurofins). After sequence analysis and evidence for genome editing at the cell population level, the cells were passaged a second time and a small proportion was seeded at very low density (10-fold serial dilutions) on a 48-well plate. After 4 weeks, clonal cell patches were identified and examined under a fluorescent microscope (Zeiss Axio Observer Z1, Oberkochen, Germany fluorescent inverted microscope). mEGFP-deficient (non-fluorescent) clones were selected, detached mechanically by gently scraping with pipette tip while aspirating at the same time and sub-cultured in separate 25cm$^2$ flasks. Three mEGFP-deficient clones were isolated, propagated and analysed for mutation at the *ifih1* locus as described above.

## Viral isolates and infections

Six viral isolates were used in this study. The Snakehead rhabdovirus (SHRV) isolate 10.23.91 was initially obtained from [32] and was propagated in EPC cells as previously described [33, 34]. The Salmon AlphaVirus subtype 2 (SAV2) cell-culture adapted isolate P42P [35] was

propagated on BFCl17 cells [36]. The Infectious pancreatic necrosis virus (IPNV) isolate 31.75, and the Infectious Hematopoietic Necrosis Virus (IHNV) isolate 25.70, were propagated in EPC cells as described by [37] and [38], respectively. The betanodavirus (NNV) isolates, SJ93Nag (a reference SJNNV strain [39]), SpDl_IAusc965.09 (a RGNNV isolate obtained from sea bass [40]), and SpSs_IAusc160.03 (a reassortant RGNNV/SJNNV isolated from Senegalese sole [41]), were propagated on E11 cells as described by [40].

**SHRV and SAV infections.** *ifih1⁻ᐟ⁻* cell lines obtained were plated into 6-well plates (approx. 400,000 cells/well) and let grow to confluency in 5 ml culture medium per well at 20˚C ($2.5.10^6$ cells per well). The SHRV or SAV2 inoculum was then added at 20˚C at a MOI of 5 or 14˚C at a MOI of 1, respectively, and the cells harvested after 3, 4, 7, or 8 days post infection (dpi).

**NNV infections.** *ifih1⁻ᐟ⁻* cell lines obtained were plated into 24-well plates at 20˚C to reach confluency and inoculated with the three NNV isolates again at 20˚C and at a MOI of 0.1 and harvested after 1, 2 or 3 dpi.

In all cases un-infected wells were kept in parallel in the same format.

## Measurement of transcript levels for viral genes and ISGs (*ifn1* and *irf3*) by quantitative RT-PCR (qPCR)

Transcript levels were quantified by relative qPCR using beta-actin as the endogenous reference gene, following the procedure described by [42]. Validation of each pair of DNA oligos was carried out by verification of a single peak melting curve post-amplification, measurement of qPCR efficiencies by establishment of a calibration curve, and by sequencing of the qPCR product. The latter step was carried out by pooling 6 qPCR products with low Cq values and purification and sequencing of the pooled products as described above. The results informed on the specificity of qPCR amplification and the potential cross-reactivity of DNA oligonucleotides between several paralogue host genes with high sequence homology. Only data generated using oligos complying with all steps of quality control, in particular generating only pure sequencing chromatograms, are presented. The *ifn1* gene analysed is the *ifna3* orthologue of the secreted isoform described in rainbow trout [43].

## SHRV and SAV infection experiments

RNA was isolated using the RNeasy Mini kit (Qiagen) with the cells collected in RLT buffer with 1% v/v β-mercaptoethanol (Merck-Sigma), stored at -80˚C until processing and homogenised using Qiashredder (Qiagen). cDNA was synthesised from 1.5 μg RNA eluate using the iScript Advanced cDNA synthesis kit for RT-qPCR according to the manufacturer's instructions (Biorad). qPCR amplification was carried out on a Biorad CFX Connect thermocycler using 300 nM each DNA oligo listed in Table 2 and the Sso Advanced Universal SYBR® Green supermix according to the manufacturer's instructions (Biorad).

## NNV infection experiments

Total RNA was purified with the E.Z.N.A. total RNA Kit I (Omega Bio-Teck) following manufacturer guidelines. After treatment with DNase (DNase I, Roche), RNA (1 μg) was reverse-transcribed with the Transcriptor First Strand cDNA synthesis kit (Roche). qPCR was conducted in a LightCycler 96 Thermocycler and the Fast Start Essential DNA Green Master Mix (Roche), using cDNA generated from 50 ng RNA. Primers used are shown in Table 2. Viral gene expression was quantified by real-time PCR as detailed by [40, 44].

## Cytopathic effect (CPE) analysis in infected cells

The CHSE-EC *ifih1*$^{-/-}$ cell lines were seeded in 96-well plates at a density of $7x10^4$ cells/well in L-15 medium (Gibco) supplemented with 2% decomplemented fetal bovine serum (FBS; Eurobio; inactivation of complement by heating at 56˚C for 30 min) and penicillin (100 U/mL)-streptomycin (100 µg/mL) (BioValley) and incubated overnight at 20˚C. The next day the cells were infected with 10-fold serial dilutions of either IPNV, stock at $2.2x10^8$ pfu/ml, SAV2, stock at $1x10^8$ pfu/ml, SHRV, or IHNV, stock at $1.1x10^7$ pfu/ml. The first well was infected at different initial multiplicity of infection (MOI) depending on the virus (IPNV: MOI 0.005; SAV2: MOI 0.2; SHRV: MOI 7.7; IHNV: MOI 0.3) and the following wells were infected with 10-fold serial dilutions from the first inoculum. After 7 to 12 days of infection depending on the virus used, the cells were fixed with 3.7% formaldehyde for 1h and stained with 0.5% crystal violet for 30 min. CPE was not analysed for NNV.

## Statistical analyses

Viral gene and ISG expression levels were compared.

**SHRV and SAV experiments**–comparisons were carried out using Student t-test on the log-transformed data (Fold changes). Post-hoc false discovery adjustment was carried out using the Benjamini-Hochberg p-value correction method [45].

**NNV experiments**—mean values were statistically analysed with the GraphPad Prism 6 software (GraphPad Software, Inc. La Jolla, USA) by two-way analysis of variance (ANOVA). Differences of $p<0.05$ were considered statistically significant.

# Results

The phylogenetic (Fig 1) and tblastn analyses showed that the *ifih1* gene is present as a single copy in the Chinook salmon genome assembly Otsh_v2.0 (GCA_018296145.1), and is also shown by previous comparative analysis across salmonids [46]. CHSE-EC cell cultures transfected with the mix of sgRNA/Cas9 complexes targeting *ifih1* and *megfp* genes contained genome-edited cells as evidenced by the Sanger chromatogram analysis justifying proceeding to the cell cloning stage. Several clones originating from single cells isolated from the CHSE-EC transfected population, mutated at the *mEGFP* locus as evidenced by loss of fluorescence, were isolated and sequenced to determine mutation at the *ifih1* locus. Clones C1 and C2 each showed homozygous mutations (ie, same mutation on both haplotypes).A 2-nucleotide deletion in the *ifih1* gene for C1 led to a frameshift generating a premature stop codon at position 24, and a single nucleotide substitution for C2 lead to a non-conservative substitution of an Arginine for a Serine at position 20 (Fig 2). A third clone did not show any mutation at the *ifih1* locus and was kept as a wild-type control cell line (WT). The cell lines were named MDA5C1 and MDA5C2, stored in liquid nitrogen and expanded for further characterisation of their phenotypes.

There were no significant differences between the MDA5C1, MDA5C2 and WT cell lines in terms of basal expression levels for the host ISG genes analysed and no differences were observed between MDA5C2 and WT cells lines in relation with gene expression following viral infection (S1 Fig).

These cell lines were infected with SHRV, SAV or NNV viruses for analysis of the induction of a number of viral and host genes. During infection with SHRV, the replication levels, as measured by RT-qPCR for the viral gene encoding the G protein, were not different between the cell lines MDA5C1 and MDA5C2 (Fig 3A) and did not show significant replication between the time points analysed (3 and 7 dpi). Between the two time points, the fold change for the *shrv g* was 0.60±0.25 and 0.61±0.25 for the cell lines MDA5C1 and MDA5C2, respectively. The infection with SHRV however induced *ifn1* and *irf3* genes significantly in MDA5C2

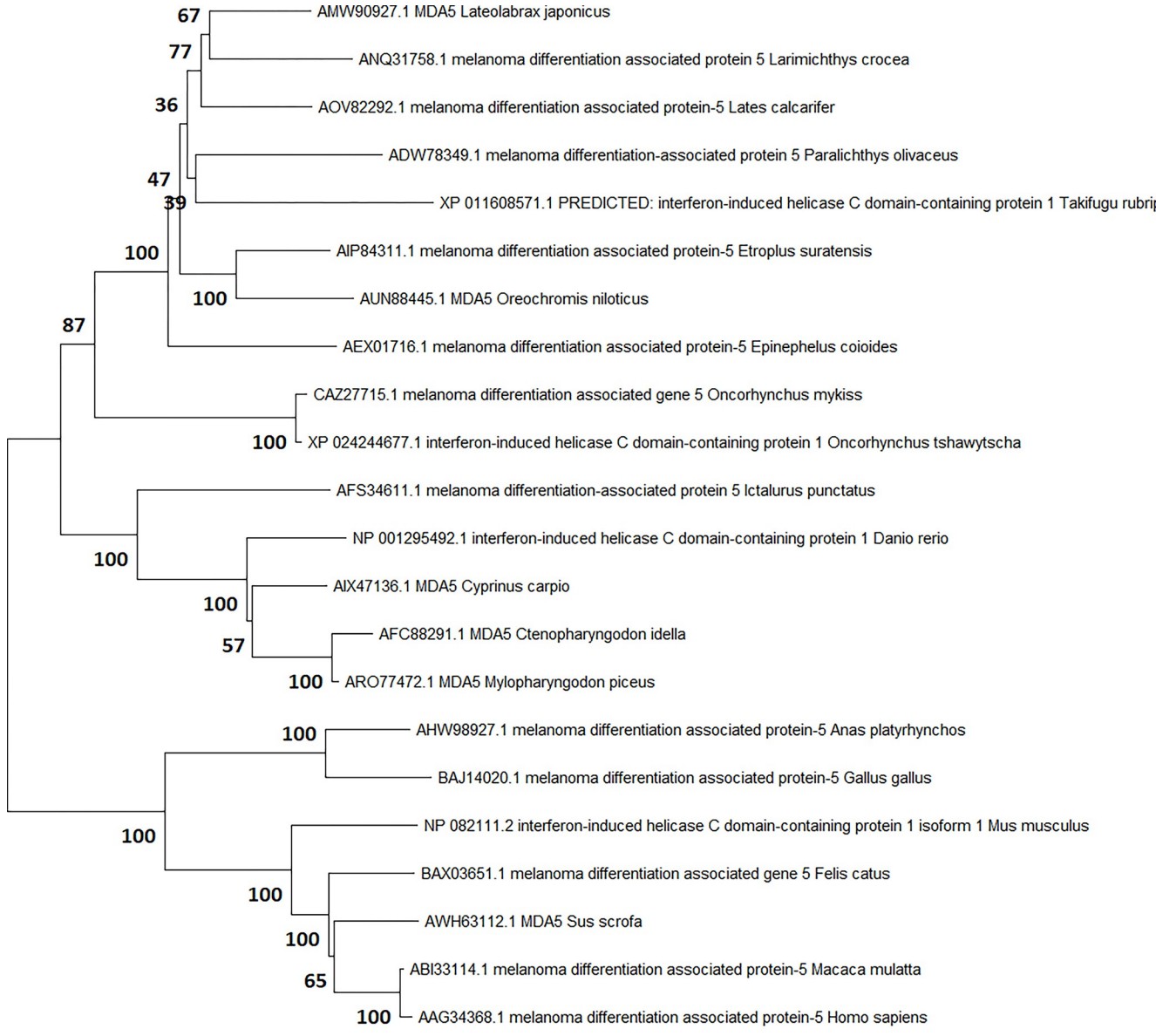

**Fig 1. Phylogenetic Maximum Likelihood tree with 500 bootstraps.** The tree with the highest log likelihood (-22104.94) is shown. The percentage of trees in which the associated taxa clustered together is shown next to the branches. Initial tree(s) for the heuristic search were obtained automatically by applying Neighbor-Join and BioNJ algorithms to a matrix of pairwise distances estimated using the JTT model, and then selecting the topology with superior log likelihood value. This analysis involved 22 amino acid sequences. There were a total of 1,060 positions in the final dataset. Tree labels shows both accession numbers and species.

compared to in MDA5C1. *ifn1* fold changes at 3 dpi were 1.62±0.20 and 13.29±3.42 for MDA5C1 and MDA5C2 cell lines, respectively (Fig 3B). At 7 dpi they were 22.04±6.20 and 123.15±6.59 for MDA5C1 and MDA5C2 cell lines, respectively. Similar results were found for *irf3* (Fig 3B).

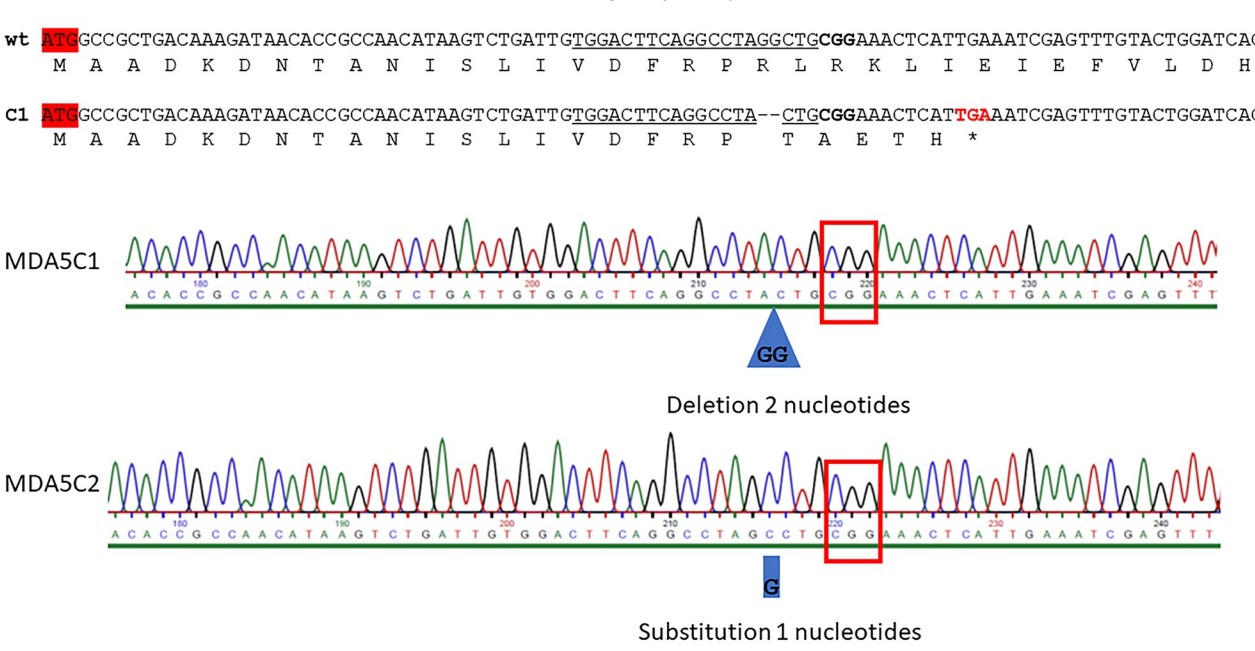

**Fig 2. Genotype of MDA5C1 and MDA5C2 cell lines.** Sequence corresponding to the first exon of the locus LOC112225192 with corresponding amino acid sequence. The clone MDA5C1 had a deletion of 2 nucleotides resulting in a frameshift and a premature stop codon. MDA5C2 had a substitution resulting in the point mutation R20S.

There was no evidence for SAV2 replication in the MDA5C2 cell line as the average Cq values from the two timepoints sampled evolved from 18.75±0.50 at 4 dpi to 21.16±0.85 at 8 dpi corresponding to a fold change decrease of 0.95 (Fig 4A). There was, however, evidence of significant viral replication (p<0.05) in MDA5C1 cells where the average Cq decreased from 19.3 ±0.78 at 4 dpi to 18.74±0.51 at 8 dpi, corresponding to a fold change increase of 4.9 (p<0.05; Fig 4A). There was a significant difference (p<0.01) between the *ifn1* induction levels between the two cell lines, with MDA5C1 having a reduced induction level of *ifn1* compared to MDA5C2 (Fig 4B). The levels of *irf3* induction were, however, not different between the two cell lines (Fig 4C).

Neither the MDA5C1 nor the MDA5C2 cell lines infected with the SJNNV isolate showed significant replication between dpi 1 and dpi 3 (Fig 5A). There was no increase in *irf3* levels at any of the time points analyzed (Fig 5A). There was a difference, however, between the two cell lines after infection with the RGNNV or the rearranged RG/SJNNV isolates (Fig 5B and 5C), with a significant replication between dpi 1 and dpi 3 in MDA5C2 but not in MDA5C1 cell line. The *irf3* transcription level was also increased only in the MDA5C2 cell line.

No notable differences between the MDA5C1 and MDA5C2 cell lines could be detected with respect to the cytopathic effect induced by the SHRV, SAV2, IPNV or IHNV isolates (Fig 6).

## Discussion

In order to understand the contribution of MDA5 in fish to the early detection of different viruses, the initiation of IFN1 induction and the subsequent antiviral resistance, we generated a clonal cell line *ifih1*[-/-], MDA5C1, in which the unique copy of the *ifih1* gene was invalidated by CRISPR/Cas9-based genome editing. The knockout (KO) of *ifih1* resulted in a clear

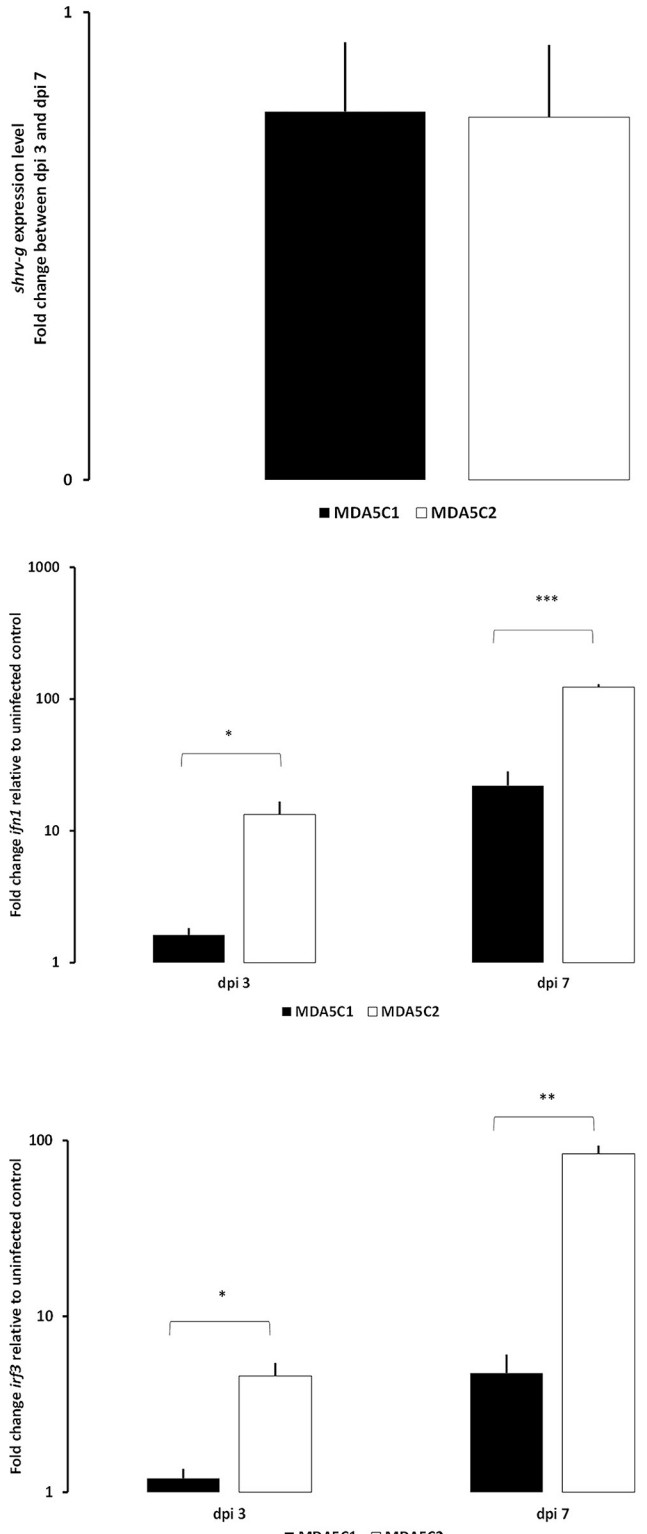

**Fig 3. . Gene expression levels of *shrv g*, *ifn1* and *irf3* in MDA5C1 and MDA5C2 cell lines.** Gene expression levels of *shrv g* (A), *ifn1* (B) and *irf3* (C) in MDA5C1 and MDA5C2 cell lines following 3 and 7 days of infection with SHRV at 20˚C. Data represent the average fold change (N = 3) + standard deviation relative to uninfected controls (B-C) or between dpi 3 and dpi 7 (A). ** = p<0.01, *** = p<0.001.

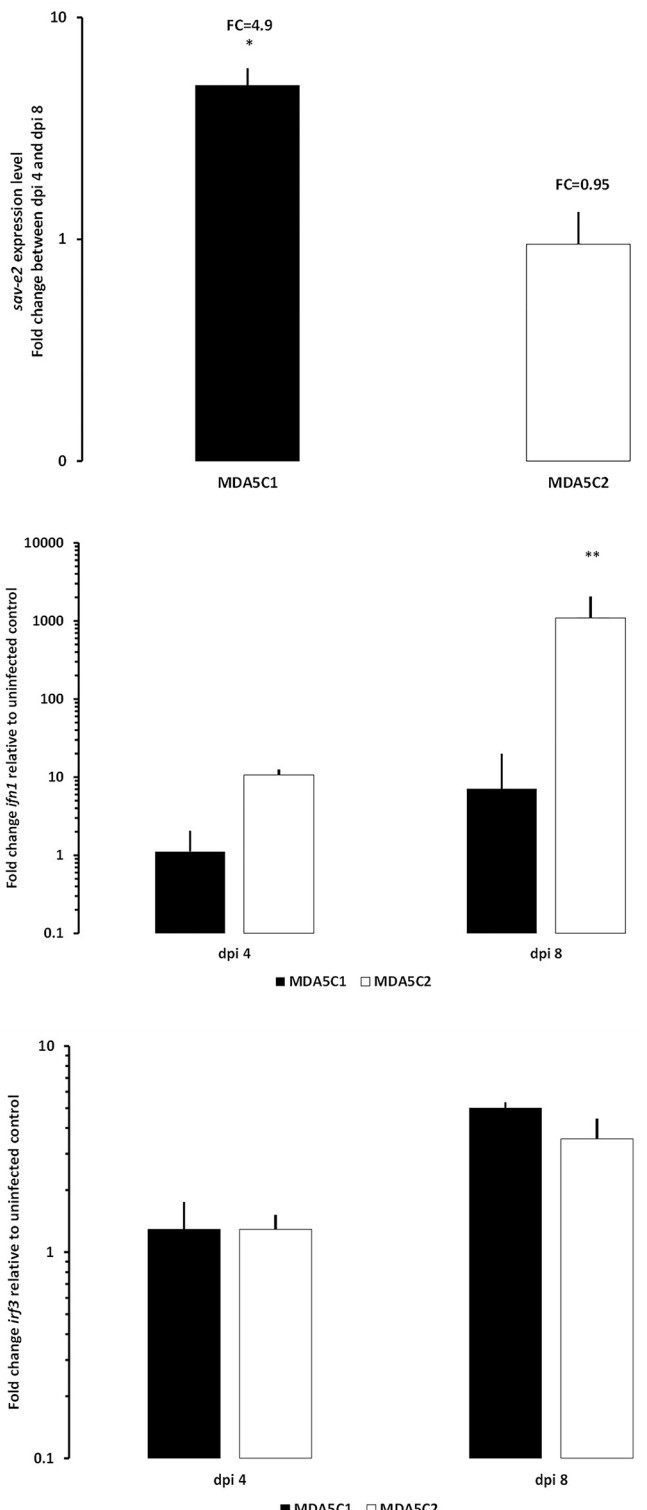

**Fig 4. Expression level of the *sav-e2, ifn1 and irf3* genes in MDA5C1 and MDA5C2 cell lines.** Expression level of the *sav-e2* (A), *ifn1* (B) and *irf3* (C) genes in MDA5C1 and MDA5C2 cells following Salmon Alphavirus 2 (SAV2) infection between dpi 4 and dpi 8. Data represent the fold changes between dpi 4 and dpi 8 (A) or between infected and non-infected cells (B).

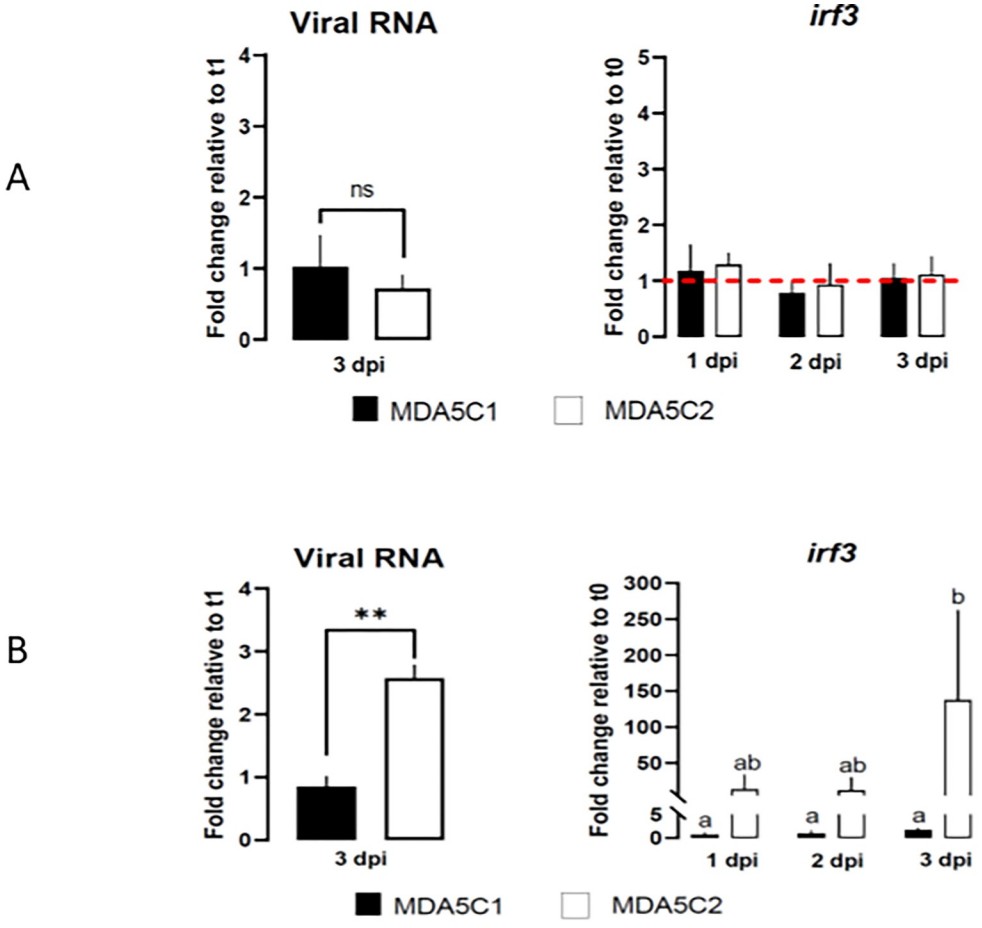

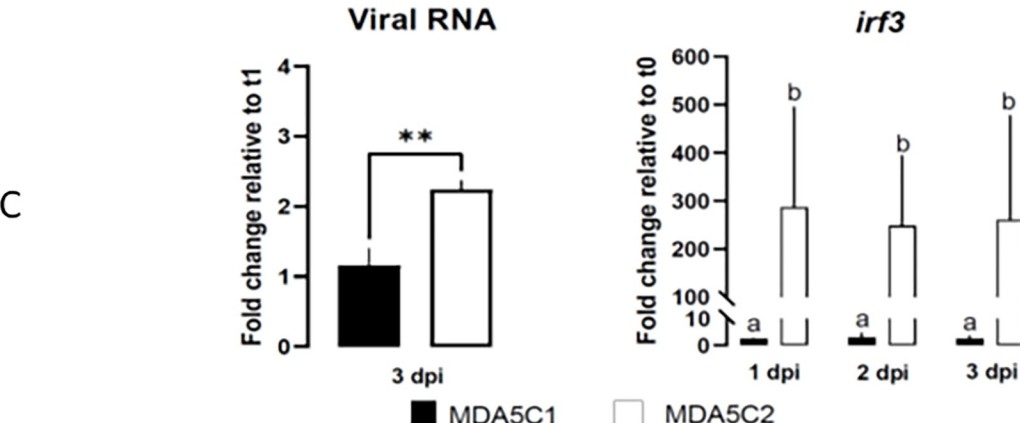

**Fig 5. Viral and host gene expression in MDA5C1 and MDA5C2 cell lines after Nervous Necrosis Virus (NNV) infection.** Replication levels and host interferon gene expression level expressed, respectively, as fold change between dpi1 and dpi3 for the viral gene (left) and relative quantification of *irf3* (right) in MDA5C1 and MDA5C2 cells infected with the NNV isolates: SJNNV (A); RGNNV (B) and RG/SJNNV (C). Data represent the average (N = 3) + standard deviation. Different letters indicate significant differences (p < 0.05).

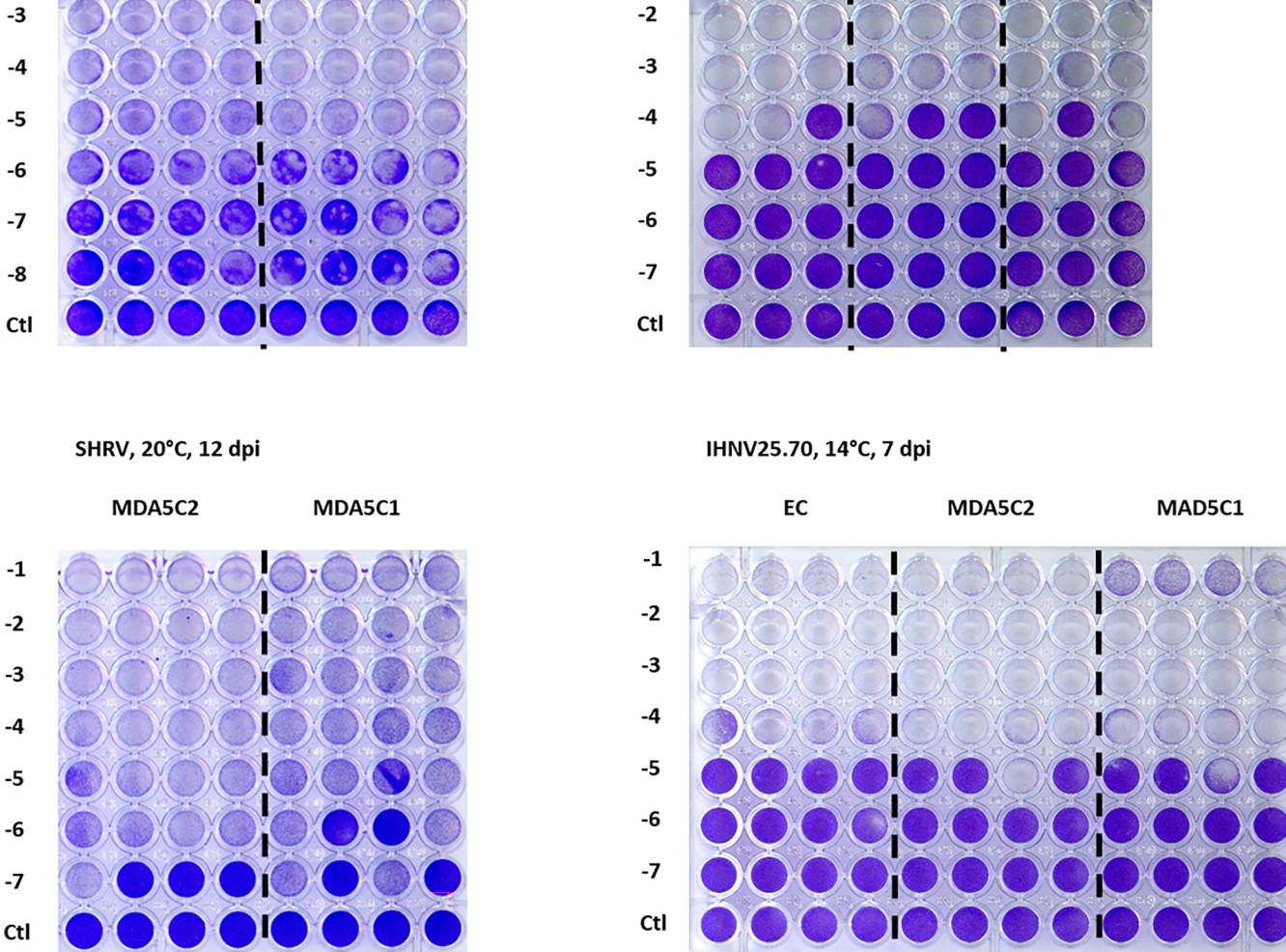

**Fig 6. Cytopathic effect (CPE) in EC, MDA5C1 and MDA5C2 cell lines following viral infection.** Comparison of virulence between MAD5C1, MDA5C2 and the parental CHSE-EC (EC) cell lines after infection with Infectious Pancreatic Necrosis Virus (IPNV), Snakehead Rhabdovirus (SHRV), Salmon Alphavirus 2 (SAV2) or Infectious Haemorrhagic Necrotic Virus (IHNV).

impairment of the induction of *ifn1* and the canonical ISG *irf3* upon infection with SHRV, a negative single strand RNA virus, or with SAV, a positive single strand RNA virus. This study is the first to establish an *in vitro ifih1* knockout fish cell line model, all other *ifih1* KO cell lines originating from mammalian models, apart from a single chicken fibroblastic *ifih1*[-/-] cell line [47]. Previously, only transient knockdown of *ifih1* in fish was achieved using siRNA [15], with uncertainty as to total efficacy.

The parental cell line CHSE-EC [31] had a sustained level of basal expression of the *ifih1* gene which was significantly induced by viral infection or stimulation with recombinant type I interferon [48].

In fish, MDA5 function was previously investigated using gain of function carried out with plasmid-based artificial overexpression of the *ifih1* gene in cell lines. It resulted in the

continuous induction of IFN1 activity and in an increase in resistance against viral infection (Rainbow trout/VHSV; [10]; Japanese flounder/VHSV-HIRRV-IPNV; [9] Zebrafish/SVCV; [12]; grouper/SGIV-RGNNV; [14]; black carp/SVCV-GCRV; [49]). In some of these studies, however, the overexpression of the *ifih1* gene was able to induce the *ifn1* gene promoter without any ligand, as well also observed for overexpression of the *rig-I* gene [50]. In mammalian models this property was linked to autoimmune diseases where MDA5 overexpression resulted in a chronic elevated level of IFN1 [51]. In fact, MDA5 seems to bind both cellular and viral RNAs, but is only activated by the later in healthy contexts [3]. Therefore, characterisation of gene function by knockout may introduce less artefacts than overexpression approaches.

Loss of function studies on mda5, either by gene knockout or knockdown, have been performed in several avian or mammalian models. The induction of *ifn1* gene by poly I:C was drastically reduced but not fully abolished in dendritic cells and macrophage primary cells isolated from *ifih1*$^{-/-}$ mice compared with those isolated from WT mice [52]. In that study, the lack of MDA5 impaired the cytokine response in peritoneal macrophages and dendritic cells to EncephaloMyoCarditis Virus (EMCV), a picornavirus with a positive ssRNA genome [52]. Similar results were obtained by [53] suggesting a critical role for MDA5 in the detection of picornavirus. MDA5 knockout has been found to be more associated with permissiveness for positive RNA viruses [54]. Replication in other categories of viruses was less affected by the absence of MDA5 in comparison to other PRRs. In the present study the effect of *ifih1* knockout was tested on the ability to activate the IFN1 pathway and to generate CPE after infection with a collection of fish viruses belonging to different categories: positive single stranded RNA (SAV2, Nodavirus), negative single stranded RNA (SHRV, IHNV) or double stranded RNA (IPNV). Surprisingly, even though the IFN1 signalling was clearly impaired, there was no clear differences in the ability for the cells to support virus infection.

A number of gene knockdown experiments were carried out in fish cell lines in an attempt to identify the consequences of a reduced *ifih1* function. Knockdown of the *ifih1* gene using siRNA in Seaperch cells increased the replication ability of NNV [15]. The increase in titre remained however moderate and below one log with no visible evidence of CPE development. *ifih1 in vivo* knockdown in Japanese flounder blocked the induction of ISGs by poly I:C and resulted in an increase in viral load after infection with megalocytivirus [55]. Overall, from the data published in fish, it is not clear as to what extent knockdown was achieved and whether the phenotype obtained was directly associated with it.

In mammalian *in vitro* models, silencing of the *mda5* gene significantly decreased IFN1-β production but only increased rotavirus titres in infected intestinal epithelial cells by less than a log difference [56]. *ifih1* single knockout mice however were shown to be highly susceptible to infection with PICV, a positive bi-segmented RNA virus [57], suggesting that the phenotype of *ifih1* mutation can be much more severe at the scale of the whole organism.

A dominant negative zebrafish mutant for *ifih1* was generated by [58] and showed an increased susceptibility to SHRV and inability to induce *ifn*Φ1 gene upon SHRV infection. In line with the present study, the mutant fish did not exhibit a complete susceptibility to SHRV suggesting other RLRs or PRRs to be involved in the IFN1 induction. We therefore decided to evaluate the effect of MDA5 loss of function on infection with SHRV. Even though we did not observe any significant replication of SHRV in our CHSE-derived cell lines, we demonstrated that the infection resulted in a strong IFN1 induction in WT cells which was impaired by the absence of a functional *ifih1* gene in the MDA5C1 cell line. In contrast with Gabor's study [58], however, the knock-out of the *ifih1* gene did not allow for replication of SHRV. It is possible that SHRV presence is detected in a MDA5-independent manner, potentially involving other PRRs, and/or an antiviral state established via an IFN1-independent pathway.

The decrease in IFN1 activity in the MDA5C1 cells was not sufficient to provide the viruses tested with an advantage compared with the WT cell line. This reflects the complexity of the interaction between IFN1, ISGs and viral propagation in a cell monolayer [59].

A *ifih1* KO quail fibroblast cell line using CRISPR/Cas9 genome editing was generated in 2021 [60]. In this study, the phenotype obtained was unexpected as the MDA5 deficient cell line was still able to induce the *ifn* gene at a high level when stimulated with poly I:C or when infected with influenza virus. Lee et al., (2020) [61] using a similar approach in epithelial chicken cells demonstrated that MDA5 is an absolute requirement for the detection of synthetic dsRNA such as poly I:C in the absence of RIG-I, which has been lost in this species. In this mutant model the replication of Avian Orthoavulavirus 1 was enhanced [47].

In the present study, the activation of the IFN1 signalling pathway was decreased in the absence of MDA5 following infection with nodavirus, SHRV or SAV2 but not completely abolished suggesting that other sensors are able to detect viral PAMPs for these viruses. The IFN1 activation was determined by measuring the transcript levels of the *ifn1* or *irf3* genes. Surprisingly, the reduced IFN1 induction was not associated with an increase in viral gene replication, except in the case of SAV2, in which a moderate rise was measured by qPCR. This agrees with the Mahesh et al study [60] on quail fibroblast cells, where *ifih1* knockout affected IFN1 induction but not the level of replication of influenza virus. However, only in double knocked-down *rig-I ifih1* fibroblastic mouse cells the production of IFN-β following infection with alphaviruses was abolished [62]. It is also consistent with the data reported from a collection of human cell lines in which single key genes of the IFN pathway, including *ifih1*, had been disrupted. These cell lines showed decreases in the expression of networks of genes but no effect on viral replication [19]. Results from Yin et al [63] follow the same direction where transient *ifih1* gene knock-down using siRNA demonstrated that the *ifn1* gene induction following SARS Cov2 infection is dependent on MDA5 in iPSC-Derived Airway Epithelial Cells.

Unfortunately, the CHSE cell line from which MDA5C1 is derived, is known to be not responsive to extracellular synthetic dsRNA such as poly I:C [64]. It was therefore not possible to simply evaluate by poly I:C incubation in the present study the dependency on MDA5 of dsRNA recognition.

Further characterisation of the MDA5C1 cell line will help in understanding the relative contribution of the different PRRs in virus-induced IFN1 expression. In particular, the comparison of MDA5C1 with CHSE-EC cell lines invalidated for genes encoding other members of the PRR family such as the genes *rig-I*, *lgp2-dhx58a* or *lgp2-dhx58b* (single mutants and/or double/triple mutants) and other PRRs will help to understand the intracellular detection pathways for viral PAMPs characteristic of different categories of viruses.

## Supporting information

**S1 Fig. IFN1 pathway induction in WT and MDA5C2 cell lines following SAV2 or SHRV infection.** Data represent the average fold change (N = 3) in the transcription levels of ifn1 or irf3 + standard deviation.
(TIF)

**S1 File. Raw data for Figs 3–5.** Individual fold changed used for the Figs 3 (tab Fig 3), 4 (tab Fig 4) and 5 (tab Fig 5). Data is organised as columns "CELL LINE", day post infection (DPI), treatment (TT), fold change between time points for infected samples, fold changes for *ifn1* (FC ifn1) and *irf3* (FC irf3) genes.
(XLSX)

## Acknowledgments

We are grateful to Stéphane Biacchesi and Emilie Mérour (INRAE, Jouy-en-Josas, France) for providing the SAV inoculum and to Pr. Niels Lorenzen (National Institute for Aquatic Resources, Technical University of Denmark, Denmark) for providing the SHRV isolate.

## Author Contributions

**Conceptualization:** Catherine Collins, Bertrand Collet.

**Data curation:** Catherine Collins, Lise Chaumont, Mathilde Peruzzi, Nedim Jamak, Julia Béjar, Patricia Moreno, Daniel Álvarez Torres, Bertrand Collet.

**Formal analysis:** Catherine Collins, Lise Chaumont, Mathilde Peruzzi, Julia Béjar, Patricia Moreno, Daniel Álvarez Torres, Bertrand Collet.

**Funding acquisition:** Bertrand Collet.

**Investigation:** Catherine Collins, Lise Chaumont, Nedim Jamak, Bertrand Collet.

**Methodology:** Lise Chaumont, Mathilde Peruzzi, Bertrand Collet.

**Project administration:** Bertrand Collet.

**Resources:** Bertrand Collet.

**Supervision:** Mathilde Peruzzi, Bertrand Collet.

**Validation:** Catherine Collins, Lise Chaumont, Bertrand Collet.

**Visualization:** Bertrand Collet.

**Writing – original draft:** Bertrand Collet.

**Writing – review & editing:** Catherine Collins, Lise Chaumont, Mathilde Peruzzi, Nedim Jamak, Pierre Boudinot, Julia Béjar, Patricia Moreno, Daniel Álvarez Torres, Bertrand Collet.

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
