## [Decision Letter · Decision Letter 0]

8 Aug 2024

PONE-D-24-20796Effect of a loss of the mda5/ifih1 gene on the antiviral resistance in a Chinook salmon Oncorhynchus tshawytscha cell linePLOS ONE

Dear Dr. Collet,

Thank you for submitting your manuscript to PLOS ONE. After careful consideration, we feel that it has merit but does not fully meet PLOS ONE’s publication criteria as it currently stands. Therefore, we invite you to submit a revised version of the manuscript that addresses the points raised during the review process.

 Please submit your revised manuscript by Sep 22 2024 11:59PM. If you will need more time than this to complete your revisions, please reply to this message or contact the journal office at plosone@plos.org. Please include the following items when submitting your revised manuscript:A rebuttal letter that responds to each point raised by the academic editor and reviewer(s). You should upload this letter as a separate file labeled 'Response to Reviewers'.A marked-up copy of your manuscript that highlights changes made to the original version. You should upload this as a separate file labeled 'Revised Manuscript with Track Changes'.An unmarked version of your revised paper without tracked changes. You should upload this as a separate file labeled 'Manuscript'.

We look forward to receiving your revised manuscript.

Kind regards,

Maria del Mar Ortega-Villaizan

Academic Editor

PLOS ONE

Journal Requirements:

4. Thank you for stating the following in the Acknowledgments Section of your manuscript:"We are grateful to Stéphane Biacchesi and Emilie Mérour (INRAE, Jouy-en-Josas, France) for providing the SAV inoculum and to Pr. Niels Lorenzen (National Institute for Aquatic Resources, Technical University of Denmark, Denmark) for providing the SHRV isolate. This project was funded in part by the European Union through AQUAEXCEL3.0 (Grant Agreement 871108) and AQUAFAANG (Grant Agreement 817923) and by the Research Council of Norway through the project PMCV (Project 301083). LC was a recipient of PhD funded by Virbac and the French Association for Research and Technology (ANRT) [Convention CIFRE #2020/0646] in collaboration with the Fish Infection and Immunity laboratory (INRAE, VIM, Jouy-en387 Josas, France).

Please remove any funding-related text from the manuscript and let us know how you would like to update your Funding Statement. Currently, your Funding Statement reads as follows: "The funders had no role in study design, data collection and analysis, decision to publish, or preparation of the manuscript."

5. We note that your Data Availability Statement is currently as follows: "All relevant data are within the manuscript and its Supporting Information files."

Additional Editor Comments:

The manuscript is of interest for the PLOS ONE readers, however, the authors should correct and resolve all the questions raised by the reviewers, in order to be accepted for publication.

Reviewers' comments:

Reviewer's Responses to Questions

**Comments to the Author**

1. Is the manuscript technically sound, and do the data support the conclusions?

Reviewer #1: Yes

Reviewer #2: Partly

2. Has the statistical analysis been performed appropriately and rigorously? 

Reviewer #1: Yes

Reviewer #2: Yes

3. Have the authors made all data underlying the findings in their manuscript fully available?

Reviewer #1: Yes

Reviewer #2: Yes

4. Is the manuscript presented in an intelligible fashion and written in standard English?

Reviewer #1: Yes

Reviewer #2: Yes

5. Review Comments to the Author

Reviewer #1: The present manuscript explores the role of Melanoma Differentiation-Associated protein 5

(MDA5) in antiviral defense in Chinook salmon cells, using CRISPR/Cas9 to create a cell line lacking the

ifih1 gene. The study found that while the induction of type I interferon (IFN1) was impaired in these

modified cells upon viral infection, there was no increase in cytopathic effects. These results suggest

that the cells have redundant antiviral defense mechanisms that compensate for the loss of MDA5.

Undoubtedly, a lot of work has gone into this paper and it is characterized by novelty. However,

some questions arise. Could the authors explain an alternative mechanism path?

Given that RIG-I and MDA5 target different dsRNA lengths, could this variability be affecting the discussion?

In lines 70-81, the text is overly verbose. Since other species also possess the specific gene, I strongly recommend moving this section to the supplementary material. The authors could also include other fish species (e.g.see genus: Gadus, Perka) for a comparative sequencing analysis.

In the manuscript, the authors state "data not shown" three times. They should either provide

this data in the supplementary material or omit these repeatead references to refine the text. Are there

any available results from at least the wild type analysis? Also, the position 20 is not significant on the

structure for protein functionality.

Although the maximum likelihood method is accurate for this type of phylogenetic analysis, the

authors should discuss the results more thoroughly and could consider using another model for the matrix.

Based on these facts, I suggest the conditional acceptance of this manuscript for publication in PlosOne.

Reviewer #2: The present report explores knockdown of a key receptor in PAMP recognition, contributing to antiviral immunity. Knockdown MDA5 cells were created using CRISPR/Cas9 and cells were infected with several important fish pathogens to look at differences in responses. Feedback is given in point form below.

- In all figures the knockdown cells are not compared to wildtype cells in figures. It was comment upon in the results that there were no differences between MDA5C2 and WT cells (225). This is misleading to then continue describing only the knockdown clones. This data must be shown in all figures.

- Manuscript needs to be thoroughly edited for grammar, many awkward sentences and atypical word choices (examples include the first sentence of the abstract, the abstract sentence "invalidated for the ifih1 gene", figure 3A y-axis label includes "et" instead of and)

- If cells were plated and then allowed to grow to confluence, how was MOI determined? Were cells re-counted prior to infection?

- Was beta-actin check for stability as a housekeeping gene?

- For SHRV and SAV 15ul of RNA was used, how was it confirmed that this RNA came from the same number of cells?

- NNV details are missing, how was RNA isolated?

- Clarify in line 186 how FBS was decomplemented

- Describe all abbreviations in figure legends

- Discussion should be re-written for flow, several "hanging" paragraphs are present that aren't integrated into the discussion, as an example line 289-291.

6. PLOS authors have the option to publish the peer review history of their article (what does this mean?). If published, this will include your full peer review and any attached files.

Reviewer #1: No

Reviewer #2: No

---

## [Author Response · Author response to Decision Letter 0]

6 Sep 2024

Please see the file "response to reviewers".

---

## [Decision Letter · Decision Letter 1]

17 Sep 2024

Effect of a loss of the mda5/ifih1 gene on the antiviral resistance in a Chinook salmon Oncorhynchus tshawytscha cell line

PONE-D-24-20796R1

Dear Dr. Collet,

We’re pleased to inform you that your manuscript has been judged scientifically suitable for publication and will be formally accepted for publication once it meets all outstanding technical requirements.

Kind regards,

Maria del Mar Ortega-Villaizan

Academic Editor

PLOS ONE

Additional Editor Comments (optional):

The authors have satisfactorily revised the manuscript and it is now acceptable for publication.

Reviewers' comments:

Reviewer's Responses to Questions

**Comments to the Author**

1. If the authors have adequately addressed your comments raised in a previous round of review and you feel that this manuscript is now acceptable for publication, you may indicate that here to bypass the “Comments to the Author” section, enter your conflict of interest statement in the “Confidential to Editor” section, and submit your "Accept" recommendation.

Reviewer #1: All comments have been addressed

Reviewer #2: All comments have been addressed

2. Is the manuscript technically sound, and do the data support the conclusions?

Reviewer #1: Yes

Reviewer #2: Yes

3. Has the statistical analysis been performed appropriately and rigorously? 

Reviewer #1: Yes

Reviewer #2: Yes

4. Have the authors made all data underlying the findings in their manuscript fully available?

Reviewer #1: Yes

Reviewer #2: Yes

5. Is the manuscript presented in an intelligible fashion and written in standard English?

Reviewer #1: Yes

Reviewer #2: Yes

6. Review Comments to the Author

Reviewer #1: The authors have satisfactorily addressed all previous concerns.The present form of the manuscript (PONE-D-24-20796R1) is suitable for publication in PLOS ONE .

Reviewer #2: (No Response)

7. PLOS authors have the option to publish the peer review history of their article (what does this mean?). If published, this will include your full peer review and any attached files.

Reviewer #1: No

Reviewer #2: **Yes: **Sarah Poynter

---

## [Editor Report · Acceptance letter]

24 Sep 2024

PONE-D-24-20796R1 

PLOS ONE

Dear Dr. Collet, 

I'm pleased to inform you that your manuscript has been deemed suitable for publication in PLOS ONE. Congratulations! Your manuscript is now being handed over to our production team.

Kind regards, 

on behalf of

Dr. Maria del Mar Ortega-Villaizan 

Academic Editor

PLOS ONE